# Combining Near-Infrared (NIR) Analysis and Modelling as a Fast and Reliable Method to Determine the Authenticity of Agarwood (*Aquilaria* spp.)

**Esther K. Grosskopf \*, Monique S. J. Simmonds and Christopher J. Wallis**

Royal Botanic Gardens, Kew, Richmond, Surrey TW9 3AB, UK
* Correspondence: e.grosskopf@kew.org

**Abstract:** The resinous wood produced by the *Aquilaria* and *Gyrinops* species—agarwood—is both rare and highly valuable. It is used in products from perfumes to medicines and has an estimated global market value of $32 billion. As a result, the adulteration and illegal purchasing of agarwood is widespread and of specific concern to enforcement agencies globally. Therefore, it is of interest to have a fast, reliable, and user-friendly method to confirm the authenticity of a sample of agarwood. We investigated the use of near infrared (NIR) data to develop a method that rapidly distinguished between authentic and non-authentic agarwood samples, based upon a soft independent model of class analogy (SIMCA), using software specific to the application of infrared data to material authentication. The model showed a clear distinction between the authentic and non-authentic samples. However, the small values involved led to poor automatic validation results.

**Keywords:** agarwood; infrared; SIMCA; authentication; wood; NIR

## 1. Introduction

Agarwood (also known as oudh, gaharu, chen xiang, eaglewood, aloeswood, or jinkoh) is a dark, fragrant, resinous wood formed as a response to wounding or fungal infection of trees of the genus *Aquilaria* or *Gyrinops* (Thymelaeaceae). These trees are distributed throughout southeast Asia and Indonesia. *A. malaccensis* is the primary tree species involved in agarwood production, although most of the 21 species of *Aquilaria* are capable of forming agarwood. Agarwood, and the oil extracted from it, has been valued as an ingredient in incense, perfumery, and medicine for thousands of years and this value shows no signs of decreasing; as of 2020, the global trade in agarwood is estimated to be worth $32 billion [1,2].

However, agarwood formation in a wounded tree is a slow and uncertain process. As few as 10% of mature *Aquilaria* trees contain agarwood and the material itself forms slowly, taking many years to reach the highest grade [3]. A highly unsustainable, though quick, method of searching for agarwood is to fell nearly every tree encountered [4]. This indiscriminate logging has contributed to the decline in the populations of *Aquilaria*. Thus, the inclusion of *A. malaccensis* in the International Union for Conservation of Nature (IUCN) Red List of Threatened Species. Because it is not easy to differentiate the timber between species of *Aquilaria* and *Gyrinops*, all species from these genera are listed in the Convention on International Trade in Endangered Species (CITES Appendix II) [5], which restricts the trade in these species. As the demand for agarwood continues to grow, the remaining wild populations come under increasing threat. Although efforts to artificially induce agarwood formation through the deliberate wounding of *Aquilaria* trees date back to the fourth century C.E., large-scale plantation growing and more reliable methods of agarwood induction have received increasing attention in the last few decades [6–8].

Given the value and rarity of this material, and the regulations on sourcing and trade (due to CITES listing), incentives for substitution and adulteration are high. This often

happens by doping low-grade wood, or unrelated cheap material, with resins or oils to mimic the appearance and scent of true agarwood. Traditionally, the authenticity and quality of agarwood was assessed by physical and organoleptic qualities such as density, colour, and response to burning [9]. However, many of these methods require subjective interpretation; colour assessment, in particular, can be spoofed with dyes [9]. Modern authentication frequently involves microscopic examination [10], which, in the case of agarwood, is largely unambiguous due to the rarity of its structural features [11]; although this is inappropriate for products such as oils and fine powders where no morphological structures exist. Other methods include thin-layer chromatography [12], DNA barcoding [13,14], and mass spectrometry [15–17]. Many of the mass spectrometric methods are based around the identification of 2-(2-phenylethyl)chromones and derivatives, as these are considered the characteristic compounds of true agarwood [18–20]: Naef et al. reported the presence of 16 highly oxidised chromone derivatives which are unique to this material [21]. Based on this, Lancaster and Espinoza identified certain ions that are considered diagnostic of agarwood [22].

These methods, however, all require expertise, time, and specialist equipment, normally only found in sophisticated laboratories. A fast, reliable, and user-friendly method for the authentication of agarwood could be of great value to customs and border control agencies, as well as growers, commercial buyers, and conservationists in the field looking to protect the species from over-exploitation.

Infrared analysis methods, widely used in wood identification [23,24], have previously been used to evaluate agarwood, largely focusing on comparing wood with and without the accumulation of resin that makes it valuable, in both near- and mid-infrared [25,26]. Infrared instruments are generally easy to use, robust, and often require minimal sample preparation. The issue remains that a reference spectrum of authentic agarwood is required to compare the sample spectrum to the authentic reference spectrum. This leaves the technique open to subjective interpretation and possible error. The objective of this study was to resolve this issue by developing a model based on NIR (near infrared) data that allows for a sample spectrum to be efficiently classified as authentic or non-authentic (adulterated or substituted), producing a simple and quick way to evaluate the authenticity of agarwood samples.

## 2. Materials and Methods

### 2.1. NIR and SIMCA Instrumentation

Analysis was performed on a Frontier FT-IR/NIR instrument (Perkin Elmer, Seer Green, UK), with an NIR accessory equipped with an InGaAs detector. Samples were placed directly on the measurement window, and each sample was analysed once. A total of 32 scans were collected between 8000–4000 cm$^{-1}$ at a resolution of 16 cm$^{-1}$. Modelling was carried out on the average of multiple scans in Perkin Elmer 'Assure ID' software using the soft independent modelling by class analogy (SIMCA) method.

### 2.2. Method Development

A total of 83 samples were available for this study, taken from material supplied to Royal Botanic Gardens, Kew (Kew) by the UK Border Force for identification as suspected agarwood. Sample details are presented in ESI Table S1. The most common form of these samples was chips of black wood, several centimeters long (Figure 1c). Larger pieces of wood, powders, amorphous lumps, and crystallised resins with embedded splinters were also present. These had previously been verified by experts at Kew either by microscopic morphological examination, or—where the sample was a powder and no microscopic structures existed—by LC-MS analysis based on the method of Lancaster and Espinoza [22] (see ESI for method details and ESI Figures S1 and S2 for representative LC-MS chromatograms). Based on this verification, the samples were classified as either Authentic (48 samples) or Substituted (29 samples). Furthermore, six of the samples were

found to have been doped: they showed the presence of some but not all characteristic chromones. These were classed as Adulterated.

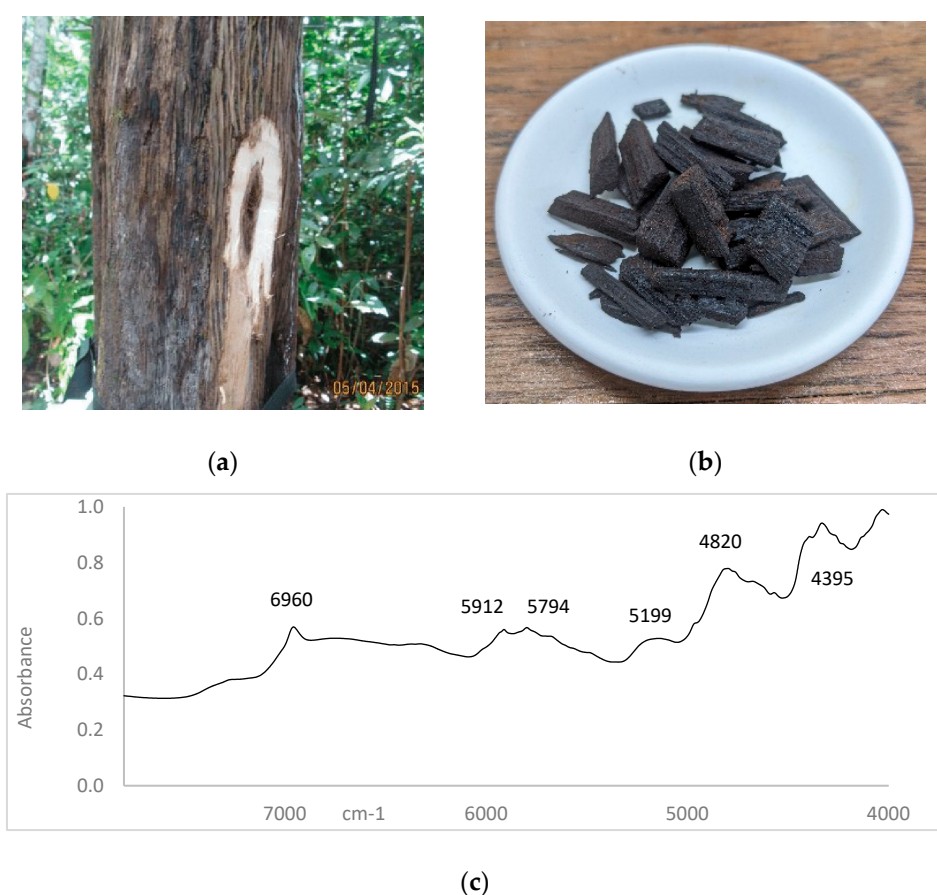

(**a**)                                                                                       (**b**)

(**c**)

**Figure 1.** (**a**) Agarwood formed inside *A. malaccensis* trunk [27]. (**b**) A representation of the most common form of agarwood samples analysed in this study. (**c**) An NIR spectrum of authenticated agarwood.

A subset of 20 samples, 10 Authentic and 10 Substituted, were used to develop the method. These samples were selected to cover as much as possible of the wide range of sample forms (chips, powder, resin). Both mid- and near-infrared instrumental methods were investigated, considering both the quality of results obtained and the practicalities of analysing a range of samples on different IR analysis accessories. Sample washing time was varied between no washing and overnight soaking. These results were used to create preliminary models. Based on these preliminary models, the optimal method was identified and the remaining 68 samples were analysed in the same way for inclusion into the model.

### 2.3. Sample Preparation for NIR Analysis

Each sample was visually inspected for signs of surface modifications or the presence of extrinsic substances. Large conglomerations of surface material were manually removed.

All samples were then washed to remove dyes and fragrances by soaking in a 90:10 mixture of ethanol:methanol (Fisher, 99.8% and 99.9%, respectively) for one hour at ambient temperature. This solvent mixture was selected to bear a close resemblance to methylated spirits, a solvent freely available and usable to anyone applying this method. The samples were rinsed with the same solvent mixture, oven-dried at 80 °C for 3 h, and cooled to ambient temperature before analysis.

## 3. Results

### 3.1. Extent of Substitution

The initial morphological or chemical analysis at Kew found 48 of the 83 samples to be Authentic, 24 Substituted, and 6 Adulterated. This represents an authenticity rate of 58%. Of the 24 Substituted samples, 5 contained no plant material at all, consisting entirely of chunks of resin or an unidentified granular material. Among these attempted substitutions were such materials as pinewood, dyed and fragranced to mimic agarwood, and corms from an unknown species thought to be a member of the Iridaceae family of plants. This demonstrates a high level of low-quality substitution entering the trade, and reinforces the need for a quick and reliable analytical method for which IR would be ideal.

### 3.2. Near-Infrared Spectra

Representative spectra of each class of sample are shown in Figure 2. There are close similarities in the spectra among the three classes; however, there is considerably more variation visible between samples within each class than between classes (see ESI for illustrating spectra Figures S3–S5). It is not a surprising finding that the chief features of the wood spectra would be similar: these refer to large structural features, such as lignins, rather than the distinctive chromones associated with quality agarwood, which exist in much lower concentrations. True agarwood and doped low-quality wood both contain these basic structures, as do many of the substitute materials, consisting as they do of wood or wood-like material.

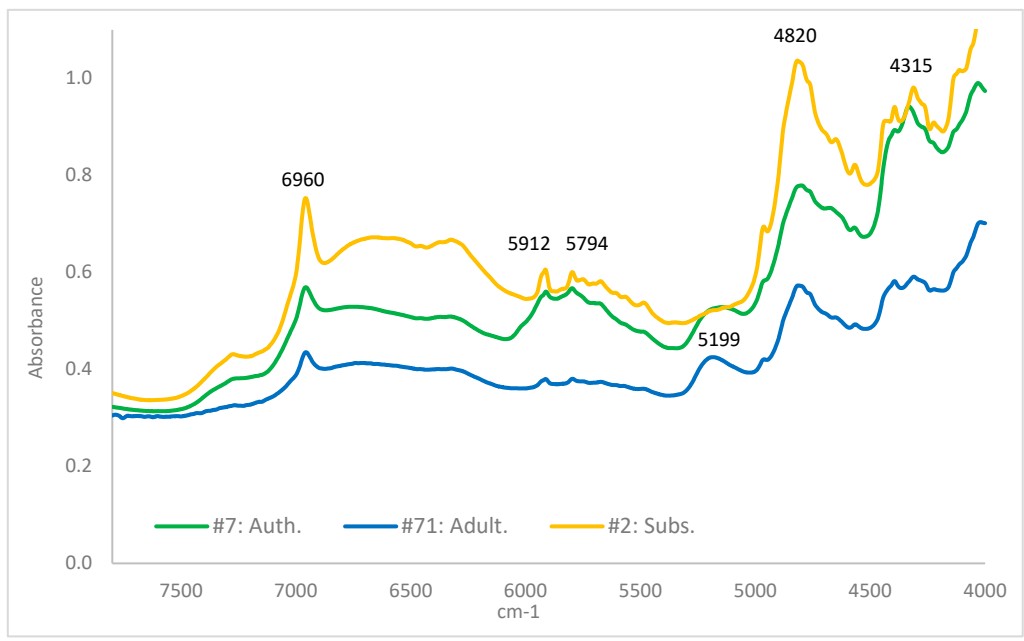

**Figure 2.** NIR spectra of one sample of each class of compound Green = Authentic. Blue = Substituted. Yellow = Adulterated.

Pre-analysis sample washing was found to be a vital preparatory step. Without it, dopants were not removed, and the model did not adequately distinguish between authentic and doped samples. However, washing for approximately 20 h (overnight) extracted many of the compounds vital to the distinction of true agarwood. Samples that had been over-washed in this way also gave rise to a model with inadequate separation between authenticity classes as the chromones on which this distinction relies were removed from the material (see ESI Figure S6 for illustrations of under- and over-washing).

## 4. Discussion

The models examined in this study were built using the SIMCA method, which is based on PCA (principal component analysis): PCA is technique which reduces the dimensionality of a dataset to render it more interpretable [28]. Broadly, in a PCA graph, the spatial relationship between two datapoints corresponds to their similarity. Such a graph showing good separation between the three authenticity groups is shown in Figure 3. Interestingly, it was found that including baseline correction as part of the data pre-processing destroyed the clear separation between each group, as the distinctions between each class of sample were drawn on very slight deviations, which were smoothed out by corrections to the baseline (see ESI Figure S7). The software used includes various baseline correction algorithms, all of which caused this effect.

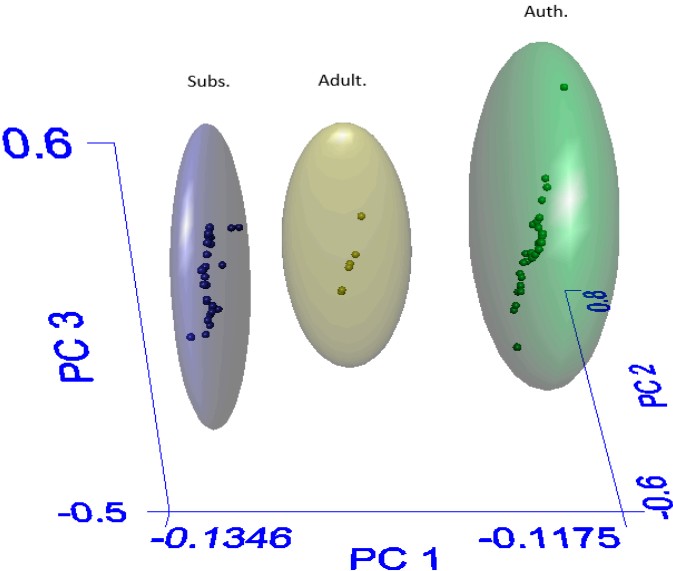

**Figure 3.** PCA graph showing clear distinctions between each authenticity class. Blue = Substituted (Subs.), yellow = Adulterated (Adult.), green = Authentic (Auth.).

The individual principal components (PCs) describe the variation in the dataset. The first principal component—PC1—is that which describes the largest amount of variation; subsequent PCs describe ever-smaller differences in the data. In this case, samples within the Substituted group (which in some cases—i.e., a corm and a chunk of resin—could be expected to have almost nothing in common with each other) do not greatly vary along the PC1 axis, but it is only along this axis that the three groups clearly differ. This is consistent with the observation that true agarwood is distinguished by the presence of chromones; it is supposed that PC1 corresponds to the presence or absence of these compounds as it is the major source of variation between the groups.

Due to these very slight differences giving rise to the variation along PC1, the range of this axis is very small: $-0.13$--$0.12$, compared to PC2 $-0.6$–0.8. This presents an obstacle to the automatic assignment of authenticity class using this model, as it is difficult to tune the model sufficiently finely to remove all ambiguity when the assignment is automated; the 'soft' designation in the SIMCA acronym refers to the fact that the model can classify samples as belonging to multiple classes. Thus, a system with low resolution can easily give rise to samples with ambiguous assignments.

The resulting model, featuring three potential authentication categories, has a best fit of recognition 95% and rejection 91% for Authentic samples (see Table 1). The confidence for the Substituted group is similarly high. However, the Adulterated group shows low specificity. Only 51% of samples that fall outside this group were successfully rejected from it, which may give rise to an unacceptable number of false positives or ambiguous assignments. This is likely to be a result of the lower resolution along the PC1 axis caused

by the inclusion of this category; the low number of samples in this group (only six) likely also contributed to the low definition of this category.

**Table 1.** Rejection and recognition for all authenticity classes in three- and two-category models. (Three categories: Authentic, Adulterated, Substituted; two categories: Authentic and Substituted).

| Class | Three Categories | | Two Categories | |
|---|---|---|---|---|
| | Recognition | Rejection | Recognition | Rejection |
| Authentic | 95% | 91% | 95% | 91% |
| Adulterated | 100% | 51% | - | - |
| Substituted | 93% | 92% | 83% | 93% |

Simplification of the model by removal of the Adulterated classification, and grouping these samples with the Substituted class to form a broader 'Non-authentic' group (see Figure 4), results in a model with a best fit of 95% recognition and 91% rejection for members of Authentic group. While the sensitivity for Substituted samples is somewhat lower when this group is widened in this way, the rejection rate remains high thus indicating that the risk of false positives is minimal. The removal of the intermediate Adulterated group increases the separation between the two remaining classes thus decreasing the likelihood of a sample being assignable to more than one group simultaneously. Therefore, it is our recommendation that the two-category method be used for evaluating agarwood samples. In real-world scenarios, such as those facing Border Force agencies or analytical laboratories, a low risk of false negatives (i.e., authentic samples falsely assigned as 'non-authentic') or ambiguous assignments will allow quick evaluations of upwards of 90% of material surveyed, dramatically reducing the number of samples that require more intensive and technically demanding analysis.

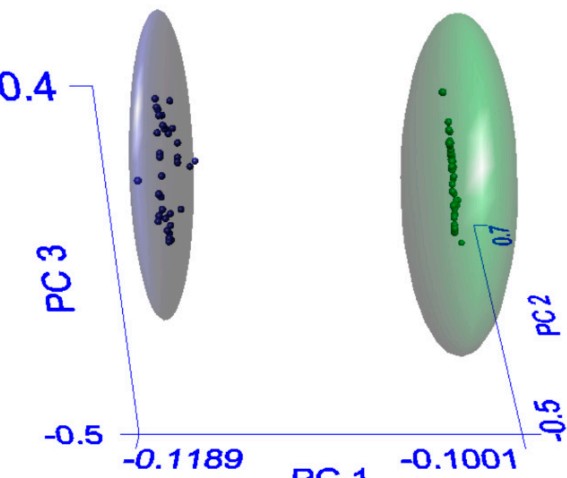

**Figure 4.** PCA graph of model simplified to Authentic and Non-authentic groups. Blue = Non-authentic, green = Authentic.

This two-category model was validated using the inbuilt validation tool in the AssureID software. In order to provide a validation set, 10 samples (5 Authentic and 5 Non-authentic) were removed from the training set (this reduction in samples available to build the model had an unavoidable negative effect upon the model itself. The selection of 10 samples represents over 10% of the samples in this study). The results of this validation were poor: only two samples (20%) were assigned to the correct class. This is due to the very small sample set and the very small differences on which the model is based. All failed samples failed because the material total distance ratio (distance from the specified material) was too high (see ESI Table S2 for full results: a value over 1.00 returns

an automatic failure as the software does not categorise the sample). In some cases, the residual distance is high, indicating that the sample contains a source of variation not yet accounted for in the model. Adding further samples to the model should overcome this issue as the categories are further resolved and more sources of variation incorporated into the model.

## 5. Conclusions

An analysis method and SIMCA model were developed to check the authenticity of agarwood samples via NIR analysis. This method and model were shown to be sufficiently sensitive to draw distinctions based on the presence or absence of key compounds found in the wood, and could correct for sample doping. When configured for a simple 'authentic/not authentic' result, the model showed a clear distinction between sample classes despite the small sample size. However, the automatic validation results were poor. The data suggest that this NIR method could enable differentiation between authentic and non-authentic materials; however, more samples are needed to expand and fully evaluate the method.

**Supplementary Materials:** The following supporting information can be downloaded at: https://www.mdpi.com/article/10.3390/analytica4020018/s1, Figure S1: Positive mode LC-MS chromatograms of three representative agarwood samples; Figure S2: Negative mode LC-MS chromatograms of three representative agarwood samples; Figure S3: Five representative NIR spectra of 'Authentic' group; Figure S4: Five representative NIR spectra of 'Substituted' group; Figure S5: Five representative NIR spectra of 'Adulterated' group; Figure S6: Model using data from washed, unwashed, and over-washed samples; Figure S7: Model with and without baseline correction; Table S1: Samples included in study; Table S2: Validation results.

**Author Contributions:** Conceptualization, C.J.W. and E.K.G.; methodology, E.K.G.; formal analysis, E.K.G.; investigation, E.K.G.; resources, M.S.J.S.; writing—original draft preparation, E.K.G.; writing—review and editing, C.J.W. and M.S.J.S.; visualization, E.K.G.; supervision, C.J.W.; project administration, C.J.W. and M.S.J.S.; funding acquisition, M.S.J.S. All authors have read and agreed to the published version of the manuscript.

**Funding:** This research received no external funding.

**Data Availability Statement:** The data presented in this study are available in the ESI.

**Acknowledgments:** The authors thank Peter Gasson and Guillermo Padilla-Gonzalez of RBG Kew for wood sample identification, through morphology and LC-MS analysis, respectively. We also thank Rhys Kelham and Peter Batey of Perkin Elmer UK for their technical assistance and support, particularly in the use and application of the modelling software.

**Conflicts of Interest:** The authors declare no conflict of interest.

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
