# Peer review of "Combining Near-Infrared (NIR) Analysis and Modelling as a Fast and Reliable Method to Determine the Authenticity of Agarwood (Aquilaria spp.)"

_analytica, doi:10.3390/analytica4020018_

Round 1
Author Response
|
This paper did not carry out sample analysis verification of the model external data set and the number of modeling samples was small, so whether the proposed method can be applied to actual production needs further research. |
Agree, and we have clarified that the robustness of the method needs further samples to fully evaluated. We have added a sentence to indicate that this is the case. |
|
Abbreviations first mentioned should be explained, such as “NIR” in line 69. |
Done. |
|
The characteristic peaks in Figure 1(c) and Figure 3 should be labelled to make the combination of images and text clear. |
Major bands have been labelled. |
|
In 2.1. NIR and SIMCA instrumentation, whether it is the average of multiple parallel scans. |
Clarified. |
|
Please check that the expression “(6)” in line 90 is complete. |
Checked, it is complete. |
|
Please check whether the NIR spectrogram in Figure 3 (c) is complete and (a), (b), (c) in Figure 3 should be marked directly below the figure |
Figure 3 updated. |
|
Suggested that the author clearly express the differences of the three types of samples according to Figure 3, so as to form a sharp contrast. |
Figure 3 updated. |
|
Table 1 shows the rejection and recognition abilities of all true and false classes in the model with principal components of Class 3 and Class 2, hoping that the authors will be more clearly and precisely in the expression of the conclusions. |
More clarification added regarding the differences between the models and why we conclude that the 2-category model is better. |
|
In paragraph 3.2. Near-infrared spectra, the author explained the importance of sample washing conditions in the analysis of the separation effect of the model and the crucial importance of distinguishing the true and false of some compounds. However, the author did not clearly clarify information about the special compounds to distinguish real from fake agarwood. |
More clarification added which we hope meets the requirements of the reviewer. |
|
Suggested to add more visual materials and data to visually show the differences under different conditions and the information of special compounds, so as to make the paper more convincing. |
Method development models and full sample data are presented in the ESI as they would take up too much space to be included in the manuscript. |
|
Your manuscript needs to be carefully edited by someone with expertise in English technical editing, paying particular attention to English grammar, spelling, and sentence structure so that the objectives and results of the research are clear to the reader. |
We have undertaken a critical review of the paper and hope we have identified the grammatical errors. Statement of objectives was added to the end of section 1. |
Reviewer 2 Report
Line 167-168, Generally, PC1 is bigger than PC2, please check.
For adulterated, there are only 6 samples, containing no plant material at all. My suggestion is that you eliminate this category sample because it will be much more complicated when the number of these samples increases, and you will not obtain a correct result and conclusion.
Line 158-161, your data (NIR spectra) does not support "that chromones are the distinguishing chemicals of agarwood and consequently, it is supposed that PC1 corresponds to the presence or absence of these characteristic compounds. "
Line 161-163, the sentence, “This also explains why baseline correction removes the distinctions along this axis: chromones exist only in low concentrations and will only give rise to very small bands, easily smoothed away by such correction” is not rigorous.
For a model, its performance should be validated by a new sample set. In this work, a validation set was missing. My suggestion is to split the samples into a calibration set for modeling and a test set for validation.
Author Response
|
Line 167-168, Generally, PC1 is bigger than PC2, please check. |
Have double checked the data, this is correct. PC1 has greater weighting but a smaller absolute value. |
|
For adulterated, there are only 6 samples, containing no plant material at all. My suggestion is that you eliminate this category sample because it will be much more complicated when the number of these samples increases, and you will not obtain a correct result and conclusion. |
There are six samples in the ‘adulterated’ category but all of them consisted of plant material. Problems caused by the inclusion of this category into this model are discussed in section 4; we conclude that these samples are better included into a broader ‘non-authentic’ category in future. |
|
Line 158-161, your data (NIR spectra) does not support "that chromones are the distinguishing chemicals of agarwood and consequently, it is supposed that PC1 corresponds to the presence or absence of these characteristic compounds. " |
Sentence clarified: we know that chromones are what differentiates agarwood from other woods, and PC1 is the major axis of variation between authentic and non-authentic samples in the study, therefore we suppose that PC1 is associated with chromones. |
|
Line 161-163, the sentence, “This also explains why baseline correction removes the distinctions along this axis: chromones exist only in low concentrations and will only give rise to very small bands, easily smoothed away by such correction” is not rigorous. |
This sentence was left over by mistake from an older version of the manuscript and has been removed. |
|
For a model, its performance should be validated by a new sample set. In this work, a validation set was missing. My suggestion is to split the samples into a calibration set for modeling and a test set for validation. |
Samples were split as suggested and validation carried out. |
Reviewer 3 Report
The manuscript presented for review raises an interesting topic. It is part of the current trends in the search for fast and environmentally friendly analytical methods combining methods of analytical chemistry with chemometrics.
Below are my comments and questions:
1. In section 2.1. the authors state that the measurement of the spectrum was carried out in the range of 10,000-4,000 cm-1. The results show a range of 8,000-4,000. The NIR range is 12,500-4,000, it should be clarified. In 2.1. the purity of the solvents used or who produced them was not specified.
2. There is no specific reference (Table S1, Figure S4...) in the main text to information in the Supplementary file.
3. "A subset of 15 samples, 5 from each authenticity class, were used in method development" - There are probably 20 points on the graphs in the Supplementary file, certainly more than the 15, where these discrepancies come from.
4. What exactly did the measured sample look like? Was any special attachment used? Why only 1 sample for each raw material? Was the measurement repeated e.g. 3 times for a given sample?
5. The information presented in Figure 2 is simple enough that there is no need to present it as a graph - the same information is in the text.
6. Figure 3 - for the spectra comparison, it is better to present them in one coordinate system.
7. The chapter on the classification model raises the most doubts. It needs significant improvements. One gets the impression that the model was built using the PCA method, which is an unsupervised technique. However, soft independent modeling by class analogy (SIMCA) based on PCA was used. The lack of any model validation makes it useless.
8. The abstract is too short and does not present the research results.
Author Response
|
In section 2.1. the authors state that the measurement of the spectrum was carried out in the range of 10,000-4,000 cm-1. The results show a range of 8,000-4,000. The NIR range is 12,500-4,000, it should be clarified. In 2.1. the purity of the solvents used or who produced them was not specified. |
Corrected sample scan range. No bands appeared above 8000cm-1.
Solvent purity and source added. |
|
There is no specific reference (Table S1, Figure S4...) in the main text to information in the Supplementary file. |
References added. |
|
"A subset of 15 samples, 5 from each authenticity class, were used in method development" - There are probably 20 points on the graphs in the Supplementary file, certainly more than the 15, where these discrepancies come from. |
Author’s error – the initial 15 samples were used to establish which IR accessory was the most practical to handle all different morphologies of sample. Method development was carried out on 20 samples from 2 categories. Section 2.2 has been updated accordingly. |
|
What exactly did the measured sample look like? Was any special attachment used? Why only 1 sample for each raw material? Was the measurement repeated e.g. 3 times for a given sample? |
More information added on the forms of the samples and how they were measured. |
|
The information presented in Figure 2 is simple enough that there is no need to present it as a graph - the same information is in the text. |
Figure 2 removed. |
|
Figure 3 - for the spectra comparison, it is better to present them in one coordinate system. |
Figure updated. |
|
The chapter on the classification model raises the most doubts. It needs significant improvements. One gets the impression that the model was built using the PCA method, which is an unsupervised technique. However, soft independent modeling by class analogy (SIMCA) based on PCA was used. The lack of any model validation makes it useless. |
Clarified that SIMCA based on PCA was used.
Validation performed and discussed. |
|
The abstract is too short and does not present the research results. |
Results have been added to abstract. |
Round 2
Reviewer 2 Report
All issues have been addressed except for t one:
Line 186-187, “Due to these very slight differences giving rise to the variation along PC1, the range of this axis is very small: -0.13 – -0.12, compared to PC2 -0.6 – 0.8”. I do not think that the PC1 range was smaller than the PC2 range. What was the chemometrics software used? I suggest that you try others.
Author Response
The software used was Perkin Elmer AssureID.
PC1: -0.13 - -0.12 is a difference of 0.01, compared to PC2: -0.6 - 0.8, difference of 1.4.
Reviewer 3 Report
Thank you very much for considering my suggestions. The current version of the manuscript is suitable for publication in the journal Analytica.
Author Response
Thank you for your kind response!